# Assessing the Level of Knowledge, Implementation Practices, and Use of Digital Applications for the Optimal Adoption of the Nutrition Care Process in Greece

**DOI:** 10.3390/nu16111716

**Published:** 2024-05-31

**Authors:** Lydia Chrysoula, Emmanouela Magriplis, Michael Chourdakis, Kalliopi Anna Poulia

**Affiliations:** 1Laboratory of Hygiene, Social & Preventive Medicine and Medical Statistics, School of Medicine, Faculty of Health Sciences, Aristotle University of Thessaloniki, 54621 Thessaloniki, Greece; lyd.chrysoula@gmail.com (L.C.); mhourd@gapps.auth.gr (M.C.); 2Laboratory of Dietetics & Quality of Life, Department of Food Science and Human Nutrition, Agricultural University of Athens, Iera Odos 75, 11855 Athens, Greece; emagriplis@eatsmart.gr

**Keywords:** nutrition care process, digital tools, dietetic practice, nutrition care

## Abstract

The level of NCP implementation varies across countries due to differences identified in major components of health systems such as infrastructures, legislation, training, and cultural diversities. Dietitians in Greece receive sufficient training in the implementation of the NCP as part of their main studies; however, the level of awareness and adoption of the NCP model is still quite low, with limited information on the potential barriers. The primary aim of this study was to gain a deeper understanding of the perspectives of Greek dietitians on the NCP and the use of digital tools. An online survey was created and distributed through the platform “SurveyMonkey version 4.1.1”. The overall structure of the questionnaire was modeled according to the validated NCP/NCPT INIS Tool. A total of 279 subjects were included in this study, and 192 were aware of the NCP tool. The most important challenges for the implementation of the NCP included communication with other healthcare professionals (68.2%), provision of appropriate care (33.9%), and insufficient access to continuous education (29.2%). Of the 192 participants who knew the NCP, 81.3% reported using digital applications for the collection and assessment of health data, while 18.8% indicated that they did not utilize such tools. No relationship was found between the use of digital applications by dietitians, NCP knowledge, and demographic characteristics. Our findings highlight the need for targeted educational interventions and appropriate application of standardized protocols by Greek dietitians in daily practice. National Dietetic Associations should provide sufficient guidance on digital tool utilization in facilitating patient data management and enhancing NCP implementation.

## 1. Introduction

Nutritional care is one the most significant elements of a healthcare system, since it is involved in every aspect of health, including disease prevention and management. The European Society of Clinical Nutrition and Metabolism (ESPEN) has defined nutritional care as “an overarching term to describe the form of nutrition, nutrient delivery and the system of education that is required for meal service or to treat any nutrition-related condition in both preventive nutrition and clinical nutrition” [1]. The Nutrition Care Process (NCP) model along with the Nutrition Care Process Terminology (NCPT) system are considered valuable profession-specific tools for dietetic professionals, facilitating a standardized strategy for performing all essential activities of their expertise within a logical framework to improve the cost efficiency of nutrition care. Therefore, dietitians should have a deep understanding of the NCP model, adequate knowledge of each stage, and sufficient confidence to implement it in various health settings such as hospitals, private clinics, nursing homes, and community health centers [2]. 

The level of NCP implementation varies across countries due to differences identified in major components of health systems such as infrastructures, workforce, leadership practices, legislation, appropriate delivery of training, and cultural diversity. The NCP model is endorsed by national and international scientific associations, including the European Federation of the Associations of Dietitians (EFAD) [3]. Evidence deriving from both quantitative and qualitative studies has shown that the application of the NCP model can positively impact the delivery of nutrition care and the quality of services within healthcare systems across the globe [4,5,6]. Australia, the United Kingdom, Canada, the Netherlands, Denmark, France, Germany, Norway, and Switzerland are among the countries that use the NCP model by ADA or their own developed NCP tool without any legal requirements, while in Austria, the implementation of the NCP is regulated by law for healthcare professionals. Even though there are multiple differences identified within all suggested nutrition care models, all share common ground on the methodological process, allowing for the development of a consolidated international model [7]. 

In Greece, there are no implementation guidelines or official recommendations to support the implementation of the NCP model. Given the multiple challenges created within the Greek healthcare system as a result of the economic crisis of 2010, there are still several functional implications closely related to the overall reductions in health expenditures and restrictions in primary healthcare services, including access to nutritional care. It was recorded that a significant proportion of the Greek population modified their dietary habits due to financial constraints, leading to an increase in the development of chronic diseases and malnutrition risks [8]. The appropriate application of the NCP by dietetic professionals in such circumstances is crucial since it provides a structured framework for interventions focused on tackling food insecurity and malnutrition—both disease- and non-disease-related—improving eating behaviors within the patient’s financial constraints and optimizing their health status [9,10].

Currently, most undergraduate courses in the United States of America (U.S.A.) and Europe include the NCP model in their curriculum [11]. Nevertheless, data obtained from national surveys, qualitative studies, and audits have demonstrated that the level of knowledge of the NCP model and the degree of implementation varies across different countries [12,13]. The most common barriers related to the implementation of the NCP and the delivery of nutritional care include insufficient knowledge of the model, lack of awareness by other health providers, inadequate emphasis and education during undergraduate and postgraduate studies, poor documentation practices, limited educational resources, and regulatory issues within healthcare systems [14,15].

In Greece, five universities offer an accredited bachelor’s degree in dietetics and nutrition science. All undergraduate courses teach modules focused on the NCP and provide students with a minimum 24-week placement in Greek hospitals. Even though dietetic practitioners in Greece receive sufficient training in the implementation of the NCP as part of their main studies, it seems that they are not familiar with this approach and, consequently, they do not use it in daily practice. In a national survey conducted in 2017, out of the 135 Greek dietitians who participated in the study, only 46% knew the NCP and almost 39% performed all NCP steps, while the obstacles that prevented dietitians from utilizing this approach in daily practice were not further explored [13]. This could be explained by the fact that, in Greece, (a) there is no official Greek licensing statute or ethical framework authorized by the Greek healthcare system for protecting the title of dietitian; (b) there is not an established regulatory body to make legal decisions for or verify the continuous educational credentials of dietetic professionals, posing many questions and concerns as to whether professionals have received their education from accredited institutions [16]. In addition, despite the recent efforts by the Greek government to develop an electronic health record system (EHR-S) for both hospital and private care settings, which was officially activated in January 2020 [17], the establishment of an electronic NCP record system has not been proposed yet.

Identifying the potential barriers encountered by nutrition professionals in Greece regarding the adoption of the NCP tool in clinical and non-clinical settings would provide valuable insights into the development of strategies and interventions for improving their education level and optimizing the utilization of digital tools in daily practice. Therefore, building upon previous research, the primary aim of this study was to assess the knowledge, implementation practices, and the use of digital tools among Greek dietitians to promote the implementation of the NCP model. 

## 2. Materials and Methods

This survey received approval by the Committee for Bioethics and Ethics, School of Medicine, Faculty of Health Sciences, of Aristotle University of Thessaloniki, Greece, on the 22 of December 2022, with protocol number 68.

### 2.1. Survey Design and Development

The Strengthening the Reporting of Observational Studies in Epidemiology (STROBE) checklist for cross-sectional studies was used to report the methods and results [15]. For this observational study, an online survey was created and distributed through the platform “SurveyMonkey version 4.1.1” based on specific guidelines on survey construction and data management [18]. The overall structure of the questionnaire was modeled according to the validated International NCP/NCPT Implementation Survey (INIS) Tool by Lövestam et al., 2019 [19]. The survey instrument was published in the Greek language and contained 32 closed- and open-ended questions grouped into 3 domains, namely “Demographic Characteristics”, “Level of knowledge and implementation of the NCP tool”, and “Use of digital tools for the optimal implementation of the NCP”. 

### 2.2. Sample Size Estimation

It is estimated that there are more than 1000 registered dietetic professionals in Greece [20]. However, the exact number of registered dietitians in Greece is unknown, and information can be accessed only by the Greek Public Health Directorate, Department of Health Services, and by Health Professions of the Competent Region of Registration. Therefore, the RaoSoft Inc. (Seattle, WA, USA) sample size calculator was used for the estimation of the required sample size, with a margin of error at 5% and a confidence level (CI) of 95% to ensure the results are statistically significant while keeping the study sample size manageable [21].

### 2.3. Inclusion and Exclusion Criteria

Registered or accredited dietitians (1) working in Greece in multiple settings, including hospitals, universities, the private sector, companies, industry, consultation services, and public administration (2) holding an accredited bachelor’s degree from any country (3) and having a valid practice license issued by the official authority of the country in which the university diploma was obtained and verified by the corresponding Greek authorities were eligible for participation. Individuals without a bachelor’s degree in dietetics or undergraduate students in relevant university courses were excluded from this study. 

### 2.4. Survey Dissemination

The online survey was made available from 9 March to 30 May 2023 and could be accessed via computers, tablets, and smartphones. Different recruitment methods were utilized, aiming to reach as many dietetic professionals as possible working in Greece. A poster including all necessary information along with the survey link was shared via social media (Facebook, Instagram, LinkedIn) and was published in online groups for professional dietitians. Furthermore, an invitation was sent by email to the two official dietetic representative bodies in Greece, the Hellenic Dietetic Association (HDA) and the Dietitians & Nutritionists Association, to share with their members. In addition, the link was advertised in national dietetic channels, networks, and online alumni groups from all five universities in Greece offering a bachelor’s degree in dietetics and nutrition. Advertisements and email reminders were sent out after 4, 7, and 9 weeks. 

### 2.5. Pilot Survey

Before the official distribution of the questionnaire, twenty dietitians were asked to complete the survey and freely submit their feedback regarding the comprehension and neutrality of all questions. The total time of completion was estimated at 6–9 min and all comments provided during the pilot phase were used to improve the quality and clarity of the questionnaire content. According to the dietitians’ recommendations, four questions were refined for better comprehension and clarity, minor wording mistakes were corrected, and two questions were removed because they were almost identical to two other questions in the same survey section.

### 2.6. Data Analysis

All data from the survey platform were exported to Microsoft Excel 365^©^ spreadsheets. Responses were organized in contingency tables and were transformed to numeric data, presented in tables, graphs, and pie charts. IP addresses were blinded to ensure confidentiality and maintain adherence to data protection regulations. Descriptive statistics were used for the demographic characteristics of responders. Data were assessed for statistical assumptions and summary statistics were expressed as frequency (percent) for all categorical variables. Statistical analysis was performed using the SPSS version 28 for Windows (IBM Corp., Endicott, NY, USA). Pearson’s chi squared test was performed to examine association and determine the degree of independence between two categorical variables. Fisher’s exact test was used in case more than 20% of the cells had expected frequencies <5, and point-biserial correlation was used to determine the association between categorical and continuous variables. Spearman’s coefficient correlation was performed to examine the relationship between categorical variables measured on a Likert scale. A very strong correlation was determined at r_s_ = 0.90–1.00, a strong correlation at r_s_ = 0.7–0.89, a moderate correlation at r_s_ = 0.40–0.69, a weak correlation at r_s_ = 0.10–0.39, and no correlation at 0.0–0.10 [22]. Statistical significance was defined as *p*-value < 0.05.

## 3. Results


*Demographic Information*


A total of 388 dietitians accessed the survey link, with a total response rate of 65% and a total time of completion of 6 min. A total of 279 subjects were included in the study analysis (Figure 1). 

Sociodemographic information is presented in Table 1. Of the 279 participants, a total of 192 were aware of the NCP tool and most individuals were between 25 and 40 years old (*n* = 205). All details with regards to participants’ place of residency, country in which their undergraduate degree was obtained, and place of work can be found in the Appendix A. Almost half of the participants who were not aware of the NCP tool had only a bachelor’s degree, 28.5% had a postgraduate diploma, and one had a PhD. The place of residency (*p* = 0.005), educational level (*p* = 0.036), and the year of undergraduate study completion (*p* = 0.010) were significantly correlated to NCP awareness. The main areas of practice of the participants included outpatient/clients, consultation services, private companies, and research (Figure 2). Figure 3 illustrates the possible definitions of the NCP reported by participants who were not aware of the tool.

The Spearman correlation matrix between the demographic characteristics and the variables related to the level of NCP knowledge, training, and use of digital tools in dietetic practice of the 192 respondents is presented as a heat map in Figure 4. The year of undergraduate study completion and the total years of working experience along with the age group and year of undergraduate study completion showed a strong negative correlation, as expected. No relationship was found between the use of digital applications by dietitians, NCP knowledge, and demographic characteristics.

The results presented in Table 2 provide insights into the NCP-related knowledge and experiences of those participants who were aware of the model. When asked whether they had received previous training on NCP theory and implementation practices, 53.1% responded positively, while 46.9% indicated that they had not received such training. In the question regarding the degree of implementation of the NCP in their daily practice, among those who reported rarely or never implementing the NCP, the main reasons cited included a lack of prior training, limited experience, and time and financial restrictions. As for the benefits of the application of the NCP in their practice, most respondents reported easier decision-making process (43.2%) and implementation of nutrition therapy (41.1%), efficient communication with clients (32.8%), improved health outcomes for clients (54.7%), and increased client adherence to nutrition therapy (26.0%). The main challenges for the implementation of the NCP included communication with other healthcare professionals (68.2%), provision of appropriate care (33.9%), and insufficient access to continuous professional education (29.2%). 

Table 3 presents dietitians’ perspectives on the use of digital applications in dietetic practice and their attitudes towards their implementation as well as the digitalization of the NCP tool. Regarding the perceived impact of digital applications on the optimization of the adoption of the NCP, participants expressed varying degrees of agreement. The types of services provided by dietetic professionals using digital applications, the main reasons for integrating digital tools into their dietetic practice, and the application brand names can be found in Appendix A.

## 4. Discussion

The present study is an attempt to provide insight into the current status of NCP adoption in Greece and shed light on the factors influencing its implementation and the potential challenges associated with the use of digital applications. Sociodemographic characteristics, including age, countries in which undergraduate degrees were obtained, year of graduation, level of education, years of professional experience, NCP knowledge, and previous training, were analyzed in depth to better understand the components that may influence NCP awareness as well as strengthen or limit the implementation of the NCP tool. Based on our results, a higher education level and greater professional experience were strongly associated with a better understating of the NCP model, as expected. Even though a significant number of dietitians in Greece have a basic understanding of the NCP, there are notable gaps in the implementation process. In addition, although the majority of dietitians reported working in the private sector and hospitals in inpatient and outpatient care, most of them would implement the NCP occasionally or rarely during practice, and only a very small percentage reported using it frequently or very frequently, while almost one-third of the total study population was not aware of the tool. Our results are in line with the findings from a previous survey conducted by Papoutsaki et al. (2017) which also assessed the level of awareness and implementation of the NCP by Greek dietitians, in which 54% of the dietitians reported not being aware of the NCP model and only a few declared that they implement it their practice [20]. Currently, there is no information available on the quality of education or students’ perspectives on the NCP study curriculums offered by the universities nationally. Potential implications on the quality of training provided during undergraduate studies in both clinical and non-clinical settings should be assessed to identify problematic areas in the education courses providing NCP training at Greek universities. 

Furthermore, our study findings indicate that the implementation of the NCP among dietitians in Greece is suboptimal. Although dietitians recognize the importance of the NCP, several barriers hinder its effective implementation. These barriers include time constraints, inadequate training, insufficient experience, and limited resources. Addressing these barriers is crucial to facilitate the adoption of the NCP and ensure its consistent application in dietetic practice. Studies exploring potential barriers to the implementation of the NCP in hospitals in Switzerland, the Philippines, and Korea also reported similar challenges, such as lack of familiarity and knowledge of the NCP model, overwhelming workload, decreased workforce, and diversity in dietitians’ educational background as well as difficulties with the implementation process [23,24,25]. Another large multinational survey that took place in ten countries, including Greece, aimed to investigate perceived barriers to and enablers of the implementation of the NCP [23]. Lack of management and inadequate knowledge, as well as insufficient education and training on the NCP model and EHR-S, were among the most common obstacles recorded by dietetic professionals. A similar survey by O Sullivan et al. (2018) explored potential predictors and attitudes on the use of the NCPT by dietitians and demonstrated that the level of knowledge and confidence were the most common predictors for utilizing the NCPT tool [26]. To overcome such barriers, efforts should be made to enhance education and training programs through workshops and seminars on standardized NCP terminology, as well as towards the development of educational materials (e.g., online tools, books, guidelines), not only at the university level but also accessible to all nutrition professionals and healthcare practitioners involved in nutritional care. Healthcare organizations including national dietetic associations and scientific societies should invest in training programs and provide adequate resources to support dietitians in implementing the NCP and NCPT effectively [27]. Healthcare organizations should invest in training programs and provide resources to support dietitians. This support can be achieved through quality improvement projects that prioritize the NCP and its integration into hospital systems. Standardized protocols, workflows, and documentation templates can be developed to streamline the implementation process and ensure consistency. Dietitians should take the lead in raising awareness about the benefits of the NCP in improving patient outcomes and healthcare quality. Interdisciplinary team meetings, training sessions, and simulations can enhance communication and collaboration among healthcare team members. Additionally, the use of electronic health records (EHRs) and the promotion of NCPT can further optimize functionality and improve patient care [28].

Moreover, our findings are of high importance, providing an insight into the integration of the NCP in healthcare settings and guiding practitioners towards the need for the establishment and integration of comprehensive nutritional guidelines into national healthcare policies to ensure consistency in nutritional care across all healthcare settings. Given the fact that nutritional care was considered a human right by the Vienna Declaration of 2022 [29], encouraging collaboration between healthcare professionals and providing lifelong learning training and education on healthcare professionals will help to incorporate the NCP efficiently into patient care.

This research also investigated how Greek dietitians leverage digital applications to incorporate the Nutrition Care Process (NCP). Based on our results, a substantial number of dietitians use digital applications in their professional practice; however, there are notable inconsistencies in the types of applications used and their features. Dietetic practitioners reported using a variety of digital apps for various tasks including nutritional assessment, documentation of dietary intake and patient progress, goal setting, practice guidelines, and use of motivational messages. It was also shown that responders would commonly recommend the use of digital apps or platforms for recording dietary intake and anthropometric measurements. However, patient/client adherence would vary, with most of them using these apps sometimes or rarely due to a lack of information or guidance on appropriate selection. The use of digital applications to enhance the adoption of the NCP carries numerous potential advantages such as increased efficiency, better precision, and simplified record-keeping [30]. These applications can provide significant aid in the interaction and cooperation among medical professionals, and at the same time permit remote supervision and assistance for patients. Our findings demonstrate a strong consensus among respondents regarding the importance of utilizing a digital app for NCP guidance and implementation. Moreover, there was a clear consensus among participants on the importance of using a uniform digital app for NCP documentation. 

A qualitative survey conducted by Chen et al. (2017) involving 381 dietetic professionals from various countries gathered information regarding dietitians’ preferences on digital tools and features for fostering dietetic practice. Several key findings were revealed, including the need for health apps that align with evidence-based practices and the importance of accuracy, reliability, and scientific validity of information provided by these apps. Responders also highlighted the need for user-friendly apps that are customizable to cater to individual patient needs and that can be integrated within working systems [31]. Our study also sheds light on the potential challenges and concerns related to health apps.

Based on these findings, it can be concluded that similar challenges for the implementation of the NCP exist universally. These challenges mainly include a lack of familiarity with and knowledge about the model, inadequate resources and support, insufficient education and training opportunities, and healthcare team intercommunication. Addressing these barriers may be a key step towards a more efficient implementation of the NCP and the improvement of the quality of nutrition care provided in hospitals and other settings of dietetic practice. 

Our study contributes to the existing body of knowledge regarding the implementation of the NCP, while integrating the use of digital technologies. In this survey, we managed to assess multiple aspects related to the adoption of the NCP in Greece. By examining the level of knowledge, implementation practices, and the use of digital tools, our study offers a comprehensive understanding of dietitians’ attitudes and perspectives, allowing for a more complete evaluation of the factors influencing the adoption of the NCP. Last but not least, our findings can serve as a reference point for future initiatives and efforts in Greece related to the adoption of the NCP tool as well as for potential reassessments on this topic. 

Some main limitations of our study include potential biases in individuals’ responses around the level of knowledge, implementation practices, and use of digital applications that might not fully reflect their actual behaviors and experiences during practice. Furthermore, the sample size of our study might not be representative of the total target population, since the size estimation was performed based on information derived only from one similar study published in 2017. Adding to this, there were many difficulties in contacting the relevant regulatory bodies and obtaining all required records with information on the total number of dietitians with valid licenses per county. Furthermore, due to the formulation of questions as part of the questionnaire, the type of data, and the small sample size, we could not perform an explanatory factor analysis to explore the underlying factors that might have contributed to the outcomes of this study. This may limit the understanding of the broader factors influencing the results and may not capture the full complexity of the topic. Overall, taking into consideration these limitations, caution should be exercised when generalizing the study findings to the broader population of dietitians. 

## 5. Conclusions

Our study highlights the need for targeted educational interventions, standardized protocols, and increased resources to enhance the adoption of the NCP among dietitians. Greece has not yet witnessed significant developments regarding the use of the NCP model and digital technologies in the field of nutrition and dietetics. Our findings underscore the importance of digital applications in supporting the NCP process and emphasize the need for the development of evidence-based tools and guidelines. Moreover, the results of our study provide direct practical implications by informing healthcare professionals, policymakers, and other stakeholders about areas that require improvement in the adoption of NCP. Future studies should focus on investigating the barriers and facilitators to NCP implementation faced by dietitians in different healthcare settings, assessing the impact of different training initiatives on dietitian competency and confidence in implementing the NCP, and exploring how certain digital applications support NCP implementation.

## Figures and Tables

**Figure 1 nutrients-16-01716-f001:**
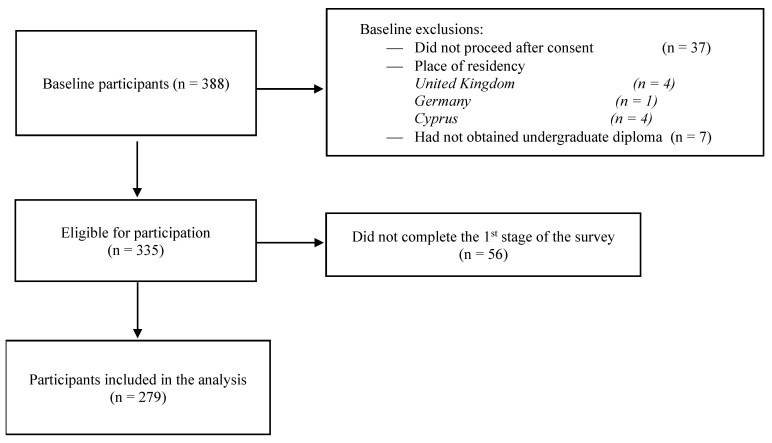
Flow diagram for the data analysis of participants.

**Figure 2 nutrients-16-01716-f002:**
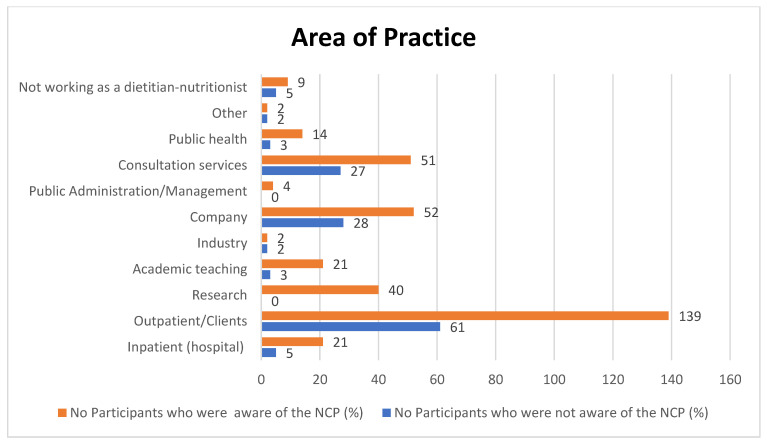
Participants’ areas of practice.

**Figure 3 nutrients-16-01716-f003:**
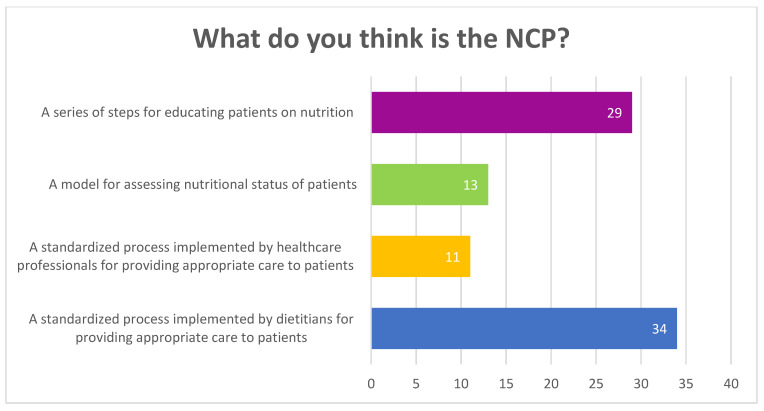
NCP perceptions of participants who were not aware of the NCP.

**Figure 4 nutrients-16-01716-f004:**
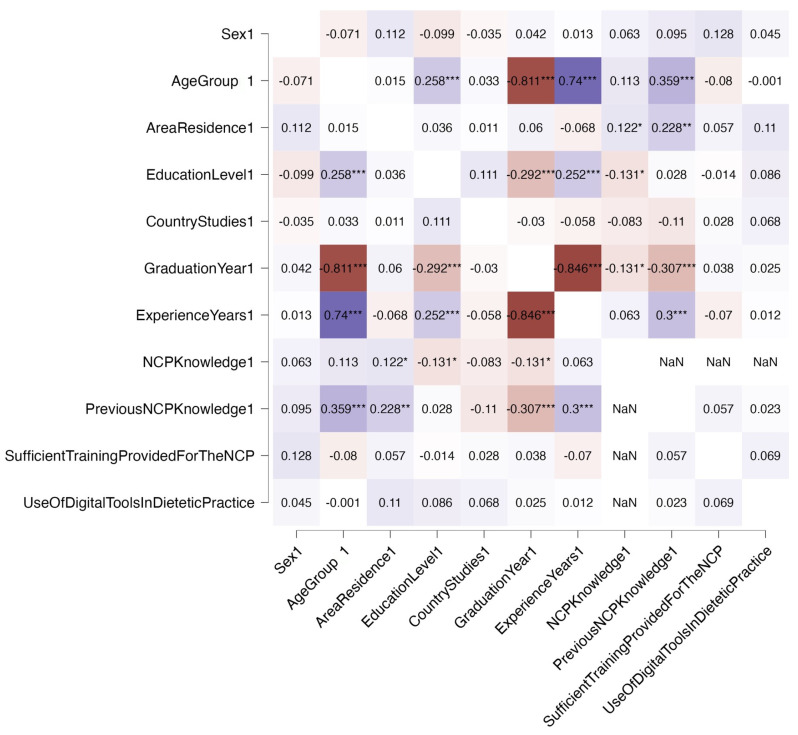
Heat map of Spearman correlation matrix between demographic characteristics and variables related to the level of NCP knowledge, training, and use of digital tools in dietetic practice. * Statistical significance *p* < 0.05. ** Statistical significance *p* < 0.01. *** Statistical significance *p* < 0.001. Colors of the boxes indicate the magnitude and direction of the correlation coefficients. Red shades represent negative correlations and purple shades represent positive correlations. Intensity of the colors reflects the strength of correlation, with darker shades representing stronger correlations. NaN = not a number.

**Table 1 nutrients-16-01716-t001:** Demographic characteristics of participants.

Variables	No Participants Who Were Not Aware of the NCP (%)	No Participants Who Were Aware of the NCP (%)	Total No of Participants (%)	*p*-Value
**Total No**	87 (31.2)	192 (68.8)	279 (100.0)	
**Sex**				
*Female*	81 (31.2)	171 (68.8)	252 (100.0)	0.290
*Male*	6 (22.2)	21 (77.8)	27 (100.0)	
**Age**				
*<25 years*	4 (14.8)	23 (85.2)	27 (100.0)	0.063
*25–30 years*	28 (27.5)	74 (72.5)	102 (100.0)	
*31–40 years*	41 (39.8)	62 (60.2)	103 (100.0)	
*41–50 years*	10 (39.8)	28 (73.7)	38 (100.0)	
*>50 years*	4 (44.6)	5 (55.6)	9 (100.0)	
**Level of Education**				
*Undergraduate Diploma*	51 (36.2)	90 (63.8)	141 (100.0)	* 0.036
*Postgraduate Diploma*	35 (28.5)	88 (71.5)	123 (100.0)	
*Doctoral Diploma*	1 (6.7)	14 (93.3)	15 (100.0)	
**Years of professional experience**				
*<1 year*	13 (24.1)	41 (75.9)	54 (100.0)	0.451
*2–4 years*	21 (29.6)	50 (70.4)	71 (100.0)	
*5–10 years*	29 (37.2)	49 (62.8)	79 (100.0)	
*>10 years*	24 (27.6)	52 (27.1)	76 (100.0)	
**Year of undergraduate study completion**				
*<2005*	11 (34.4)	21 (65.6)	32 (100.0)	* 0.010
*2005–2010*	16 (30.8)	36 (69.2)	52 (100.0)	
*2011–2015*	27 (49.0)	28 (50.9)	55 (100.0)	
*2016–2020*	23 (27.1)	62 (72.9)	85 (100.0)	
>2020	10 (18.2)	45 (81.8)	55 (100.0)	

* *p*-value < 0.05.

**Table 2 nutrients-16-01716-t002:** NCP perceptions of participants who were aware of the tool.

Variables	No Participants (%)
**Total No**	192 (100.0)
**Where did you hear about the NCP for the first time?**	
*During university studies in Greece*	141 (73.4)
*During university studies in abroad*	6 (3.1)
*During a conference/congress*	4 (2.1)
*Social media*	17 (8.9)
*Working environment*	5 (5.4)
*Scientific article/Book*	10 (5.2)
*Dietetic/Nutrition Society*	1 (0.5)
*Seminar/Webinar*	6 (3.1)
*Scientific Society*	2 (1.0)
**Have ever received previous training on NCP theory and implementation practices?**	
*Yes*	102 (53.1)
*No*	90 (46.9)
**In what degree do you think you implement the NCP?**	
*Very frequently*	14 (7.2)
*Frequently*	28 (14.6)
*Occasionally*	88 (45.8)
*Rarely*	50 (26.0)
*Never*	12 (6.25)
**What are the main reasons for not applying the NCP in daily practice?**	*n* = 62
*No prior training received*	11 (17.7)
*Not enough experience*	22 (35.5)
*Limited time*	11 (17.7)
*Difficulty with the documentation process*	6 (3.1)
*Lack of financial resources*	13 (20.9)
*Not necessary/useful*	8 (12.9)
**What are the main benefits of applying the NCP?**	
*Easy decision making on nutrition therapy*	83 (43.2)
*Easier to implement the nutrition therapy*	79 (41.1)
*Efficient communication with the client*	63 (32.8)
*Improved application of the nutrition therapy by the client*	70 (36.5)
*Improved health outcomes of your client*	105 (54.7)
*Increased adherence of your client to the nutrition therapy*	50 (26.0)
*Other ^a^*	2 (1.0)
**What are the main challenges for applying the NCP?**	
Conducting the nutritional assessment	31 (16.1)
Identify the nutritional diagnoses	42 (21.9)
Monitoring and evaluation of the nutrition care plan	52 (27.1)
Documentation	17 (8.9)
Limited communication with other healthcare professionals (doctors, psychologists, nurses, pharmacists)	131 (68.2)
Failure to provide appropriate training	65 (33.9)
Insufficient provision of professional continuous education	56 (29.2)
No challenges	5 (2.6)
**Do you feel that you have access to educational seminars for NCP in Greece?**	
Yes	23 (11.9)
No	169 (88.0)

^a^ (1) Better organization of nutrition care stages, (2) Improved communication between healthcare professionals.

**Table 3 nutrients-16-01716-t003:** Perceptions of participants who were aware of the NCP with regards to the use of digital applications and the digitalization of the NCP tool.

Variables	No Participants (%)
**Do you use digital applications for the collection, recording and assessment of patient health data?**	Total N = 192
*Yes*	156 (81.3)
*No*	36 (18.8)
**How often do you recommend to your patients/clients to use a digital app/platform to record dietary intake and anthropometric measurements?**	
*Always*	7 (3.6)
*Often*	68 (35.4)
*Sometimes*	47 (24.5)
*Rarely*	41 (21.4)
*Never*	29 (15.1)
**How often do your patients/clients use a digital app to record food intake and anthropometric measurements?**	
*Always*	1 (0.5)
*Often*	60 (31.3)
*Sometimes*	60 (31.3)
*Rarely*	53 (27.6)
*Never*	18 (9.4)
**What are the main reasons you are not using a digital app/platform in dietetic practice?**	
*Lack of information/guidance on the most appropriate digital applications suggested for dietitians*	79 (41.1)
*Do not have access to smartphone/tablet/computer*	8 (4.2)
*Digital applications are hard to use*	18 (9.4)
*Lack of resources (e.g., access to Wi-Fi)*	11 (5.7)
*Services covered by the applications do not provide support/guidance*	40 (20.8)
*Most applications require paid subscription*	55 (28.6)
*Digital applications do not add value to dietetic practice*	17 (8.9)
**To what extent do you think that the use of a digital application/platform would help to optimize the adoption of the NCP?**	
*To a great extent*	11 (5.7)
*A lot*	83 (43.2)
*Somewhat*	72 (37.5)
*Little*	23 (11.9)
*Not at all*	3 (1.6)
**How important do you think it would be for all dietitians to use a common digital app/platform as a tool for guidance and implementation of the NCP?**	
*Not important at all*	3 (1.6)
*Of little importance*	15 (7.8)
*Of Average importance*	47 (24.5)
*Very important*	94 (48.9)
*Absolutely essential*	32 (16.7)

## Data Availability

Data are contained within the article and Appendix A.

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
