# Peer review of "Assessing the Level of Knowledge, Implementation Practices, and Use of Digital Applications for the Optimal Adoption of the Nutrition Care Process in Greece"

_nutrients, 2024, doi:10.3390/nu16111716_

Round 1
Reviewer 1 Report
Comments and Suggestions for Authors
Abstract:
· The initial comments on the training received by Greek dietitians and their prevalent patient-centered approach are overly general. Linking this background directly to the specific knowledge or practice gaps addressed by this study would make better use of this section.
· The abstract cites specific statistics (e.g., 192 dietitians are familiar with the NCP, and 81.3% utilize digital tools) but lacks a detailed discussion on the implications of these numbers for practical or policy applications. Expanding on how these findings impact the adoption of the NCP in Greece would make the abstract more relevant.
· Digital tools are mentioned briefly in the abstract without detailed analysis on which tools are used or their impact on NCP implementation. This significant element of the study should be highlighted more prominently.
· The final remarks in the abstract are generic. Phrases like "highlight the need for further education" are too vague. Instead, detailed recommendations or implications based on the findings would be more meaningful.
1. Introduction:
· Providing more context about the unique healthcare challenges in Greece and how the NCP can address them would strengthen the justification for the study.
· While the usage of the NCP internationally is well-documented, the transition to the situation in Greece feels abrupt. A smoother transition would better set the stage for discussing why Greece lags behind in adopting the NCP.
· The research gap could be articulated more clearly at the beginning. It's apparent that the study aims to understand Greek dietitians' perspectives on digital tools and the NCP, but stating explicitly how little research exists in this area and the importance of bridging this gap would be beneficial.
· A 2017 survey is mentioned, but it's unclear how the current study builds upon or differs from this previous work. Clarifying this would help position the current study within a broader research context.
· The study's objectives are implied but not explicitly stated. Clearly articulating the objectives at the end of the introduction, such as "This study aims to assess the knowledge, implementation practices, and the use of digital tools among Greek dietitians to promote the NCP," would guide the reader more effectively.
2. Materials and Methods:
· The exact number of licensed dietitians in Greece is unknown, so using the RaoSoft sample size calculator to estimate the necessary sample size is appropriate. However, the rationale for the chosen margin of error and confidence level could be clearer. Discussing potential limitations due to the sample size calculation based on limited information would provide a more balanced view.
· The study's inclusion and exclusion criteria are well-defined, targeting a specific and relevant group of dietitians. Broadening the exclusion criteria to include undergraduates and those without a bachelor's degree could help address potential biases or skewness in the results.
· The comprehensive distribution strategy using emails, professional networks, and social media likely enhanced the survey's reach. However, detailing the effectiveness of each dissemination strategy would be useful. For example, reporting response rates from different channels could illuminate the most effective recruitment strategies for future studies.
· Conducting a pilot survey was a strong point of the methodology, helping to identify and address issues before the full survey implementation. Discussing specific changes made to the survey based on feedback from pilot participants would increase the process's transparency and rigor.
3. Results:
· The demographic breakdown is detailed, but a deeper analysis of how these demographics influence NCP awareness and implementation would be beneficial.
· It's noted that a strong correlation exists between NCP knowledge and factors like the year of graduation, educational level, and location, but the reasons for this correlation are not explored.
· While 53.1% of participants reportedly received NCP training, the impact of this training on actual implementation practices is not examined here.
· Financial constraints and a lack of training are cited as barriers to NCP implementation, but the broader context of the Greek healthcare system is omitted.
· It's positive that 81.3% of participants use digital applications; however, more details are needed on the types of applications used and how they specifically enhance NCP implementation.
· The section on patient/client adherence to digital tools is informative but requires more qualitative data to fully understand patient perspectives.
· No narrative explanation accompanies the correlations shown in the heat map. A detailed discussion of the key correlations would be helpful.
· Tables 2 and 3 are useful, but the text could benefit from a more detailed exploration of the implications, such as the fact that 88% of participants lacked access to educational seminars on the NCP in Greece.
4. Discussion:
· Although the section outlines barriers, it should delve deeper into the root causes of these issues. For instance, why is there no training even though NCP is included in the curriculum? What specific aspects of training are missing? Addressing these questions could lead to a more comprehensive study.
· The discussion would be enhanced by more specific suggestions for overcoming the identified barriers. Recommendations on particular educational programs, legislative changes, or resource allocations would enrich the text.
· The potential of digital applications is mentioned, but this topic could be explored more thoroughly. What specific features of digital tools are most beneficial? Are there examples of best practices from other countries that could be adopted in Greece?
· The discussion of limitations is brief. While potential biases and sample size issues are noted, exploring the implications of these limitations on the study's findings in more detail would be beneficial. How might these limitations affect the generalizability of the results?
· The section could be improved by recommending specific areas for further research. Studies could be conducted to determine how well certain digital technologies support NCP implementation or how specific training initiatives affect dietitian competency.
· Some parts of the discussion could be clearer. For instance, the statement "Our findings demonstrate a strong consensus among respondents regarding the importance of utilizing a digital app for NCP guidance and implementation" could be supported with more specific data or quotes from respondents to illustrate this consensus.
· More details on the broader implications of the findings for the Greek healthcare system and potentially other similar healthcare settings should be provided. How might these findings inform the integration of nutritional care into broader health programs or national healthcare policy?
Comments on the Quality of English LanguageThe text is well-constructed and clear and generally adheres to academic standards of grammar, vocabulary, and stylistic coherence, making it comprehensible without significant effort. There are occasional minor errors or awkward phrases, but these do not substantially detract from the overall readability or understanding of the content.
Reviewer 2 Report
Comments and Suggestions for Authors
This great article examines whether dietary guidelines are actually followed by the health care workers themselves, as well as important hurdles to be taken. The workout is well done and the tables & figures are very informative. I do have some questions:
- Table 1 & Figure 4: apparently, only recently graduated dieticians are well informed about the NCP. Can this be explained?
- The guidelines are based on those of ESPEN. However, the ASPEN guidelines might be more "accessible", which could help to resolve the research question.
Author Response
|
We would like to thank Reviewer#2 for the kind words! · Table 1 & Figure 4: apparently, only recently graduated dieticians are well informed about the NCP. Can this be explained? -According to table 1, not only recent graduates are aware of the NCP, but also a great number of individuals aged 25-40 years old as well as individuals with 1-10 years of professional experience. |